# Structure and Immune Function of Afferent Lymphatics and Their Mechanistic Contribution to Dendritic Cell and T Cell Trafficking

**DOI:** 10.3390/cells10051269

**Published:** 2021-05-20

**Authors:** Jorge Arasa, Victor Collado-Diaz, Cornelia Halin

**Affiliations:** 1Vaccine Research Unit, Fundación para el Fomento de la Investigación Sanitaria y Biomédica de la Comunidad Valenciana (Fisabio), 46020 Valencia, Spain; jorge.arasa@fisabio.es; 2Institute of Pharmaceutical Sciences, ETH Zurich, CH-8093 Zurich, Switzerland; victor.collado@pharma.ethz.ch

**Keywords:** dendritic cell, T cell, afferent lymphatic vessel, migration, leukocyte, trafficking, lymphatic endothelial cell, immunity

## Abstract

Afferent lymphatic vessels (LVs) mediate the transport of antigen and leukocytes to draining lymph nodes (dLNs), thereby serving as immunologic communication highways between peripheral tissues and LNs. The main cell types migrating via this route are antigen-presenting dendritic cells (DCs) and antigen-experienced T cells. While DC migration is important for maintenance of tolerance and for induction of protective immunity, T cell migration through afferent LVs contributes to immune surveillance. In recent years, great progress has been made in elucidating the mechanisms of lymphatic migration. Specifically, time-lapse imaging has revealed that, upon entry into capillaries, both DCs and T cells are not simply flushed away with the lymph flow, but actively crawl and patrol and even interact with each other in this compartment. Detachment and passive transport to the dLN only takes place once the cells have reached the downstream, contracting collecting vessel segments. In this review, we describe how the anatomy of the lymphatic network supports leukocyte trafficking and provide updated knowledge regarding the cellular and molecular mechanisms responsible for lymphatic migration of DCs and T cells. In addition, we discuss the relevance of DC and T cell migration through afferent LVs and its presumed implications on immunity.

## 1. Introduction

The lymphatic system consists of central or primary lymphoid organs (bone marrow and thymus), peripheral or secondary lymphoid organs (SLOs), such as spleen and lymph nodes (LNs), and an extensive network of lymphatic vessels (LVs) that penetrate most tissues of the body. It is essential for maintaining tissue homeostasis by draining fluids and macromolecules and in taking up dietary fats from the intestine [1,2,3]. Moreover, the lymphatic system constitutes a crucial compartment of the immune system, since it transports antigen and leukocytes from peripheral tissues to draining LNs (dLNs) and from there back into the blood circulation [1,4,5]. The first step in which antigen and leukocytes are drained from the periphery is mediated by afferent LVs. In recent years, increasing evidence has revealed an important role of these vessels in regulating and shaping adaptive immunity and immune surveillance in general [4,6]. Therefore, unraveling the mechanisms by which migratory cells migrate into and within afferent LVs will be vital to better understand how the immune system functions and to develop improved therapeutic approaches.

LVs were described for the first time in in the early 17th century [7], but it was not until the end of the 20th century when the discovery of lymphatic-specific markers, such as the vascular endothelial growth factor receptor-3 (VEGFR-3) [8], the lymphatic vessel endothelial hyaluronan receptor-1 (LYVE-1) [9], the mucin type-1 protein podoplanin [10] and the lymphatic-specific transcription factor PROX-1 [11], opened the way for molecular and cellular research in the field of lymphatic biology. Although these markers are also expressed by other cell types (e.g., podoplanin by glomerular podocytes [10], LYVE-1 by macrophages [12] or PROX-1 by muscle satellite cells [13], among others), the combination of them allows to unequivocally identify the lymphatic vasculature in different tissues and isolate lymphatic endothelial cells (LECs), e.g., transcriptomic analysis or in vitro cultivation. Moreover, the generation of fluorescent reporter mice, together with modern time-lapse imaging techniques performed ex vivo and in vivo, has greatly contributed to elucidating the mechanisms by which leukocytes migrate through afferent LVs.

In peripheral tissues, leukocyte trafficking through initial lymphatics occurs sequentially. The process is initiated with the interstitial migration of leukocytes towards afferent LVs, followed by an active intravasation and intraluminal crawling which eventually is followed by cellular detachment and passive transport to dLNs with the lymph flow [4,5]. From cannulation studies in the 1970s and more recent technical advances, such as the use of photoconvertible proteins, it is nowadays known that the main cell types migrating through afferent LVs are antigen-presenting dendritic cells (DCs) and antigen-experienced T cells [14,15,16,17,18]. Upon antigen recognition, DCs mature and migrate through afferent lymphatics to dLNs to induce adaptive immune responses that will lead to protective immunity or maintenance of tolerance [19,20]. On the other hand, T cells trafficking through afferent LVs contribute to immune surveillance and control of immune responses in peripheral tissues [21].

In this review, we first introduce the morphologic characteristics of afferent LVs and common methods used for investigating leukocyte trafficking. We then summarize current knowledge of the cellular and molecular mechanisms of DC and T cell migration through afferent lymphatics, with a particular focus on new insights gained by time-lapse and intravital microscopy (IVM). In addition, we will discuss the functional relevance of DC and T cell migration through afferent LVs for the induction and regulation of the immune response.

## 2. Characteristics of the Lymphatic Vasculature

Afferent LVs begin as blind-ended lymphatic capillaries in peripheral tissues that merge into downstream, contractile segments of the vasculature, known as lymphatic collectors, which subsequently connect with dLNs (Figure 1). Efferent LVs exit from dLNs to reach other LNs and eventually fuse, forming the thoracic duct and the right lymphatic duct that connect the lymphatic vascular system with the cardiovascular system. The prime role of LVs is to collect interstitial fluid, macromolecules and leukocytes from peripheral tissues and to return this load, which is commonly referred as lymph, through dLNs and back to blood circulation [2,3,4]. To fulfill this process, the different segments of the lymphatic vasculature have specific anatomical and morphological characteristics.

Capillary LVs are wide, thin and blind-ended structures in peripheral tissues that are constituted by partly overlapping oakleaf-shaped lymphatic endothelial cells (LECs) [22]. These cells are attached to each other with discontinuous button-like cell–cell junctions, thereby generating gates, known as flaps or primary valves, through which cells and fluids can access to the lumen. The opening of these flaps is controlled by the interstitial fluid pressure of the tissue, since capillary LECs (capLECs) are anchored to extracellular matrix (ECM) with filaments that exert force on the cells and open the flaps when the fluid pressure in the tissue increases [23,24]. Furthermore, initial lymphatics are surrounded by a thin and highly fenestrated basement membrane which further contributes to vessel permeability (Figure 1) [25].

Collecting LVs represent deeper segments of the vasculature that form part of afferent and efferent LVs. Their main function is the leakage-free transport of the lymph from peripheral tissues to dLNs, between LNs and eventually into the blood [26]. Collector LECs (colLECs) have an elongated shape and are tightly attached to each other with tight, zipper-like cell–cell junctions [22]. These zippers, together with the presence of a thick basement membrane [25] that encloses the vessel, make collectors rather impermeable. Lymphatic collectors are also surrounded by lymphatic muscle cells [27] that allow them to contract, thereby generating lymph flow [26]. Because of the presence of intraluminal valves that prevent backflow, the lymph is propagated towards dLNs (Figure 1) [26,28]. The vessel segment comprised between one valve and the following one is known as lymphangion.

The lymph, propelled by contractile afferent collectors, is released into the subcapsular sinus (SCS) of the dLN. The long-held assumption has been that soluble lymph components are primarily distributed according to their molecular size, which dictates entry into the specialized LN conduit system formed by fibroblastic reticular cells [29,30,31,32]. However, a recent study revealed that lymph-borne antibodies and large molecules that are physically unable to enter the LN via the conduit system may get transported by dynamin-dependent vesicular transcytosis across the LECs lining the SCS into the LN parenchyma [33]. The access route of leukocytes into the LN parenchyma, on the other hand, appears to primarily depend on cellular size and the cell’s ability to advance through the SCS, which represents a three-dimensional sieve formed by LECs and collagen fibers that span between the SCS ceiling and floor [34]. While antigen-presenting DCs and antigen-experienced T cells, which are rather large cells, get easily trapped in the SCS and, hence, transmigrate through the LEC layer that constitutes the SCS floor [34,35], naïve T cells, which are smaller in size, are transported along the SCS until the medullary sinuses (MS) from where they enter into the LN parenchyma [34,35]. Filtered lymph leaves the dLN at the level of the MS through an efferent collecting vessel. Considering that LNs are often arranged in chains, the efferent vessel exiting from one node can at the same time be the afferent vessel for the subsequent dLN. In this review, we will focus on leukocyte transmigration from peripheral tissues through afferent LVs, understanding afferent lymphatics as the vessels that initiate in a peripheral tissue and do not derive from an upstream-located LN.

## 3. Cellular Composition of Afferent Lymph

Cannulation studies conducted in sheep and healthy humans have revealed that the cell composition of the afferent lymph in steady-state conditions consists mainly of T lymphocytes (80–90%) [14,36,37]. Interestingly, the majority of T cells trafficking through afferent lymphatics are antigen-experienced CD4^+^ effector memory T cells (T_EMs_), which are believed to recirculate from peripheral tissues back to the blood circulation, thereby providing immune surveillance. Because naïve T cells migrate mainly between the blood and SLOs and are generally not equipped with the homing molecules required for extravasation out of blood vessels in peripheral tissues, they are only found in low numbers in afferent lymph. CD8^+^ T cells are also present in afferent lymph, albeit at smaller numbers compared to CD4^+^ T_EMs_ [38,39,40,41]. More recent studies performed in photoconvertible mice (see Section 4) have revealed that regulatory T cells (T_reg_) represent a prominent CD4^+^ T cell subset trafficking through afferent LVs in steady state (20–25%) and in inflammation (up to 50%) [38,42,43].

Besides T cells, DCs constitute the second major cell population in afferent lymph. Cannulation studies in sheep and healthy humans revealed that they represent 5–15% of all cells in afferent lymph [14,17,36,37]. Other immune cells, such as monocytes, granulocytes or B cells, are also present in afferent lymph, although in low numbers compared to T cells and DCs [36,39]. However, their numbers significantly increase in inflammation [44,45].

## 4. Methods Used to Study Leukocyte Migration through Afferent Lymphatic Vessels

The investigation of leukocyte migration through afferent lymphatics started more than 50 years ago with cannulation studies revealing the cellular composition of afferent lymph. Meanwhile, several other in vivo and in vitro methods have been established to investigate the molecular mechanisms of this process, both at the population and at the single-cell level. A summary of the techniques used is provided in the following and in Figure 2.

Considering that leukocyte migration into and within afferent lymphatics is a complex process that highly depends on the tissue environment, in vivo methods are the most relevant and translational experiments. *Cannulation of afferent lymphatics* is usually conducted in large species, such as in sheep or in humans [14,36,37,39]. However, more recently, methods for cannulating lymphatics in mice and rats have also been established [46]. In *adoptive transfer experiments*, fluorescently labeled or otherwise marked leukocytes are injected into a peripheral murine tissue and its dLN is subsequently analyzed to determine the numbers of migrated cells. Injection sites typically are the murine footpad or ear skin, since these tissues primarily drain to one LN, i.e., the popliteal or auricular LN, respectively. In addition to analyzing the dLNs, the distribution of the adoptively transferred leukocytes inside and around afferent LVs in the proximity of the injection site can be analyzed by preparing tissue whole mounts (examples in [47,48]). While the adoptive transfer setup allows for a lot of flexibility in the choice of leukocytes, e.g., wild-type (WT) and knock-out (KO) cells labeled in two different colors can be co-injected in a 1:1 ratio and their migration analyzed in a competitive manner (examples in [49,50]); an obvious disadvantage of this assay is that leukocytes frequently derive from in vitro expansion/differentiation protocols and, therefore, might not reflect the behavior of endogenous cells. Additionally, the injection procedure itself might cause damage at the injection site or the cells and passively force leukocytes into lymphatics, particularly if too large volumes are injected. In the case of DCs, *FITC or TRITC painting* experiments, which allow endogenous DC migration to be investigated, offer an attractive alternative. They rely on the DCs’ ability to continuously sample and take up material from their environment. After applying a solution containing the contact sensitizers FITC or TRITC onto the shaved mouse ear or belly skin, dLNs are typically harvested 24–72 h later and fluorescent DCs are quantified (examples in [51,52,53,54]). In analogy to FITC painting, endogenous DC migration can also be investigated by transfer of fluorescent microspheres/particles. While small particles (20 –200 nm) can reach LNs by free drainage, larger particles (500–2000 nm) typically require cellular transport [55]. By *administration of fluorescent microspheres* (*larger particles*) into the skin or into the lung, migration of particle-bearing DCs to dLNs can, therefore, be assessed in a quantitative manner [56,57,58]. FITC/TRITC painting and microparticle injection work well for investigating DC migration, but these methods fall short for T cells, which do not exert major phagocytic/endocytic activities. An elegant method to study endogenous migration of any cell type arose from the development of transgenic mice with ubiquitous expression of a photoconvertible protein, such as Kaede and Kikume [15,18]. In a *photoconversion experiment*, the skin and sometimes surgically exposed organs are illuminated with violet light, leading to the conversion of all cells in the illuminated tissue from a green to a red-fluorescent state. This allows photoconverted cells that derive from the illuminated tissue be subsequently identified in the dLN via flow cytometry [15,18,38].

While all of the above-mentioned methods investigate leukocyte migration at the population level, *time-lapse imaging in tissue explants* [48,49,59,60] and *intravital microscopy* (IVM; examples in [61,62,63,64]) allow this process to be studied at the single-cell level. The use of these methods has delivered important insights on how DCs, T cells and neutrophils approach, enter, and migrate within afferent lymphatics (discussed further in Section 5, Section 6 and Section 7). IVM to visualize lymphatic migration has been performed in the ear skin [62,65], footpad [61] or cremaster muscle [66]. It is facilitated by the use of fluorescent reporter mice to visualize lymphatics [67,68,69,70] and different leukocyte populations [64,71,72,73]. For time-lapse imaging in ear skin explants (examples are [25,49,74]), lymphatics can also be visualized with fluorescent antibodies whilst leukocytes (typically DCs) can be labeled with fluorescent dyes prior to addition to the tissue. Besides time-lapse imaging, skin explants can also be used for so-called *crawl-out assays*. These experiments rely on the fact that, once human or murine skin specimen are taken into culture, DCs in the tissue get activated and migrate towards and into LVs [75,76,77], from where they are thought to exit into the culture medium primarily via lymphatics [78,79,80].

In addition to the above-mentioned in vivo/ex vivo methods, also several functional in vitro assays are used to investigate the involvement of candidate molecules in particular migration steps. *Adhesion assays* are usually carried out in 96- or 48-well plates and evaluate involvement of adhesion molecules or chemokines in leukocyte adhesion to LEC monolayers. *Transmigration assays*, use transwell chambers and chemotactic gradients to assess leukocyte transmigration across LEC monolayers. Finally, *crawling assays* investigate the mobility and directionality of leukocytes on LECs monolayers in presence or absence of laminar flow by time-lapse imaging (examples in [59,79]). The above-mentioned in vitro studies have mostly been performed with human primary LECs and leukocytes (primarily DCs) [53,81], since the inability to isolate and culture primary murine LECs has for a long time hampered similar assays in the murine system, or limited them to the use of immortalized LECs [48,59,82]. Recently established protocols for the cultivation of primary LECs isolated from murine LNs [83] or tail skin [84] have greatly expanded the options of these assays, by, e.g., allowing to perform experiments with LECs isolated from knock-out mice [74,79].

In the following sections, we will first illustrate the general steps in leukocyte migration through afferent lymphatics and then describe in greater detail the functional relevance and molecular requirement for DCs and T cell migration, i.e., the two major cell types migrating via afferent lymphatics.

## 5. Migration through Afferent Lymphatics Occurs in a Step-Wise Manner

The skin is the organ in which lymphatic migration has been best studied so far. The main leukocytes cell types present in steady-state skin are macrophages, memory T cells, DCs and some more rare cells, such as mast cells or innate lymphoid cells [85,86]. During episodes of inflammation, the cellular composition changes, as inflammatory cells such as neutrophils, monocytes or T_EM_ are additionally recruited. These cells either remain in the tissue to execute their function, die or in the case of monocytes differentiate to become monocyte-derived DCs or macrophages. Additionally, under both steady-state and inflammatory conditions, cells constantly exit from the tissue again in a process that exclusively takes place through afferent LVs. Time-lapse imaging experiments have established that lymphatic egress occurs in a step-wise manner (Figure 3A,B).

In a first step, leukocytes approach the lymphatic capillaries by actively and, hence, slowly migrating and squeezing through the interstitial matrix (Figure 3B, panel 1). While some tissue-resident cells such as mast cells or macrophages are rather immotile, DCs and T cells migrate at average velocities of 4–5 μm/min in steady-state skin [62,65,87]. Conversely, in the course of an inflammatory response, more rapidly migrating T cells and neutrophils get recruited into the tissue [65,88]. The migration of leukocytes towards lymphatics is regulated by chemotactic guidance cues which will be further discussed in Section 6 and Section 7. Electron microscopy and time-lapse imaging studies have shown the second step in the lymphatic migration cascade, i.e., entry into the vessel, mainly occurs at the level of the lymphatic capillaries (Figure 3B, panel 2). This step involves squeezing through the thin and fenestrated BM surrounding capillaries and entering the vessel lumen through the characteristic open flaps, also called primary valves, which are formed by neighboring LECs [22,25] (Figure 3B, panel 2; and Figure 1). IVM studies have revealed that even once leukocytes have entered the vessel, they continue to actively crawl and migrate (Figure 3B, panel 3) within the capillary lumen [61,62,64,65,66]. The reason why cells are not simply flushed away, as they would in blood vessels, is the low lymph flow in this compartment (average velocities of 3–5 μm/s [89,90]), which does not support passive transport by flow. Nevertheless, the speed of intralymphatic migration is slightly faster than in the interstitial tissue, possibly because cells no longer need to navigate through the dense ECM [62,65]. Intriguingly, leukocytes do not migrate straight towards the lymphatic collectors but rather arrest or patrol for many hours within capillaries (Figure 3B, panel 3) [59,63]. A study from our group found that in the case of DCs this migration occurs in a semi-directed manner, and that only approximately 20% of DC’s net movement (approximately 1.2 μm/min) accounts for migration in the downstream direction of the dLN [59]. Once leukocytes arrive in collecting vessel segments, lymph flow increases due to the vessel contractions mediated by the surrounding LMCs. Consequently, leukocytes now start to detach and may be transported in a passive manner with the lymph flow towards the dLN (Figure 3B, panel 4). In comparison with lymphatic capillaries, lymph flow in dermal collecting vessels appears to be much higher: Performing IVM in the murine ear skin, we for instance observed a T cell that was rolling with a velocity of approx. 1 mm/min (16 μm/s) in a contracting dermal collector, suggesting that the true velocity of lymph flow must be even higher (see video S6 in [65]). In the case of large collecting vessels in the mesentery, velocities of up to several mm/s have been measured [91,92]. Considering that the speed of movement/transport in capillaries and collectors differs by several orders of magnitude, transition to flow in collectors represents a crucial step needed for the cell’s timely arrival in the dLN.

The above-mentioned step-wise migration pattern reflects the current paradigm of leukocyte migration through afferent lymphatics (Figure 3B, panels 1–4). However, there is emerging evidence that lymphatic trafficking might in some cases also occur via a different route. A recent study from our lab revealed that DC entry is not always restricted to lymphatic capillaries. Performing time-lapse imaging and in vivo migration studies we could show that under inflammatory conditions DC entry may also occur downstream of capillaries, into the adjacent lymphatic collector segments (Figure 3B, panel X) [74]. Contrary to lymphatic capillaries, lymphatic collectors are embedded in a much thicker BM and fibrillar collagen layer than capillaries [25,62], which, however, is degraded in inflammation, while concomitantly several adhesion molecules important for DC trafficking become upregulated in collector LECs (see Section 6). By directly entering into lymphatic collector segments, DCs likely avoid the slow, active migration in capillaries and manage to arrive faster in the dLNs than the bulk of DCs that have entered through capillaries [74].

In the following the functional relevance and molecules involved in DC and T cell migration through afferent lymphatics (Section 6 and Section 7, respectively) will be outlined.

## 6. Dendritic Cell Migration through Afferent Lymphatics

### 6.1. DC Types and Functional Relevance of Migration

DCs are antigen-presenting cells (APCs) that play a fundamental role as sentinels of the immune system, bridging between innate and adaptive immunity. In tissues, immature DCs display a highly branched tree-like shape, hence their name, and constantly sample their environment with their dendritic protrusions. DCs are highly specialized in capturing, processing and presenting antigen on major histocompatibility complex (MHC) molecules, to induce activation of naïve, cognate T cells. As such, they are crucial for induction of adaptive immunity in the context of infection or also vaccination. If a DC encounters antigen in presence of signs of an infection or tissue damage (i.e., in presence of so-called PAMPS (pathogen-associated molecular patterns) and DAMPS (danger-associated molecular patterns), it becomes activated and starts to mature. This leads to the upregulation of molecules related to antigen presentation, expression of cytokines and co-stimulatory molecules required for efficient T cell activation as well as trafficking molecules, which allow the DC to translocate from the peripheral tissues to dLNs, where activation of naïve T cells primarily takes place. While DC migration is strongly enhanced in inflammation [50,52], DCs also continuously travel to dLNs in the steady-state condition.

It is important to note that the DCs themselves cannot discriminate between self and foreign antigen and will indiscriminately present peptides derived from any captured proteins on their MHC molecules. Antigen taken up in steady-state conditions and in absence of an infection will, therefore, mainly be derived from self-proteins. However, in this case, DCs typically are immature/semimature and express only low levels of co-stimulatory molecules. As a consequence, cognate (autoreactive) T cells are not activated in dLNs; rather, T cell death, anergy or conversion to a tolerogenic state (i.e., a T_reg_) is induced. DC migration to dLNs under steady-state conditions is, therefore, of key importance for the control and maintenance of peripheral immunological tolerance. Accordingly, mice lacking LVs display signs of autoimmunity [93,94].

DCs in peripheral organs exist in many different subsets, which may vary between tissues [95,96,97]. In skin, Langerhans cells (LCs) represent the sole DC subset populating the epidermis, whereas the dermis hosts so-called conventional DCs (cDC1 and cDC2) as well as monocyte-derived DCs (moDCs) and plasmacytoid DCs, which get recruited into the skin inflammation. These DC subsets originate from different precursors and can be distinguished by the expression of different markers [95,96,97]. Furthermore, different DC subsets play distinct roles in the immune response and, for example, vary in their ability to crosspresent antigen, or to mediate immunostimulation or immunotolerance [95,96,97]. Besides DCs that reside in peripheral tissues and migrate via afferent lymphatics to dLNs, also resident DCs exist in SLOs. The latter derive from precursors that enter LNs via the blood vasculature and also play important roles in antigen presentation. For example, resident DCs can take up and present antigen that has arrived in dLNs via afferent lymphatics [98,99]. Moreover, they are known to take up and efficiently present antigen transported via migratory DCs to dLNs [100,101]. Resident and migratory DCs also display differences in the type of T cells they activate; for example, in viral infections, migratory DCs were shown to prime CD4^+^ but not CD8^+^ T cells in dLNs, whereas resident DCs prime CD8^+^ T cells in a delayed manner, helped by the already activated CD4^+^ T cells [102].

As mentioned, the number of DCs migrating under inflammatory conditions is considerably increased compared to the steady-state condition [50,52]. Still, the kinetics of this phenomenon are rather slow, especially compared to the much more rapidly migrating neutrophils [64,103]. This can in part by explained by the fact that, in response to a migration-inducing stimulus, DC first need to get activated and upregulate chemokine receptor CCR7, which guides their migration towards lymphatics (see below). In the case of LCs, the cells first need to exit the epidermis by transmigrating the epidermal BM to access the lymphatics-containing dermis. Consequently, their arrival in dLNs is delayed; while numbers of dermal DCs in dLNs peak 1–2 days after application of a migration-inducing stimulus, LC numbers reach maximal levels after 3–4 days [18,54]. Despite these differences, all DCs and in fact also all T cells (see Section 7) present remarkable similarities in their dependence on one key molecular pathway for lymphatic migration, i.e., the CCR7-CCL21 axis.

### 6.2. Molecular Mechanism of DC Migration

#### 6.2.1. CCR7-CCL21 Axis

The main molecules involved in DC migration towards afferent lymphatics under inflammation and steady-state conditions are the chemokine receptor CCR7 and its chemokine ligand CCL21 [50,51,104]. CCR7 is upregulated on both DCs in response to a maturation-inducing stimulus (DAMPs or PAMPs) and on semimature DCs migrating in steady-state conditions [50,105,106]. Both genetic deletion of Ccr7 and CCL21 blockade have been shown to severely impair DC migration to dLNs [51,104].

In mice, CCL21 is encoded by two genes resulting in two functional CCL21 proteins that differ in a single amino acid: CCL21-Leu and CCL21-Ser. Experiments assessing leukocyte migration in *plt^−/−^* mice, which show diminished expression of CCL21-Ser and of the other known CCR7-ligand CCL19, have revealed that CCL21-Leu is mainly produced by LECs of peripheral afferent lymphatics, whereas CCL21-Ser is mainly produced by LECs in LNs [107]. Therefore, CCL21-Ser and CCL19 drive DC migration into the LN, whereas CCL21-Leu is responsible for migration through afferent LVs [51,75,108,109]. Because we cover leukocyte migration through afferent LVs in this review, we will refer CCL21-Leu as CCL21 from now onwards.

CCL21 is secreted radially by LECs in afferent capillaries [60] and, to a lesser extent, in afferent collectors [59,74]. However, a substantial amount of CCL21 is stored intracellularly in the trans-Golgi network [60,81]. The secreted chemokine CCL21 contains a positively charged N-terminus that interacts with heparan sulfate (HS) proteoglycans present in surrounding intersticial ECM and on the LEC surface [110,111]. This interaction leads to the formation of a fixed haptotactic chemokine gradient in the interstitium that guides activated DCs towards afferent lymphatics [60]. Whole-mount immunoflourescence microscopy revealed that this immobilized peri-lymphatic CCL21 gradient radiates from lymphatic capillaries up to 90 μm into the interstitium [60]. Surprisingly, a recent study has shown that LEC-produced HS is dispensable for the formation of the mesenchymal CCL21 gradient and does not affect DC migration towards lymphatic capillaries, suggesting that the ECM surrounding lymphatic capillaries is the major site of CCL21 anchoring [112]. CCL21 stored in intracellular depots was shown to be released after exposure to inflammatory cytokines such as TNF-α [81]. Furthermore, interactions with DCs induce Ca^2+^ signaling in capillary LECs, thereby triggering the release of intracellular CCL21 and deposition of a CCL21 track for subsequently migrating DCs [113].

However, CCL21 is not only important for the interstitial guidance of activated DCs towards afferent LVs. Our group has shown that CCL21 also guides intralymphatic DCs to migrate in a down-stream-directed manner within lymphatic capillaries [59]. Specifically, time-lapse experiments performed in murine skin explants and in in vitro crawling assays using flow revealed that LEC-secreted CCL21 forms an intralymphatic gradient guiding intralymphatic DCs in their migration from lymphatic capillaries towards collectors.

While the function of CCL21 in leukocyte migration into and through afferent lymphatics is uncontested, the picture is less clear for CCL19, i.e., the second CCR7-ligand. CCL19 is produced by mature DCs and fibroblastic reticular cells and by or around high endothelial venules (HEVs) in SLOs [114]. Unlike CCL21, CCL19 does not have a positively charged C-terminal domain and, therefore, does not form an immobilized chemokine gradient. Excess of CCL19 reportedly leads to CCR7 desensitization towards CCL21 in DCs [115]. However, genetic deletion of Ccl19 had no impact on DC migration trough afferent lymphatics [116]. In contrast, studies in mice deficient in the atypical chemokine receptor 4 (ACKR4) revealed an involvement of CCL19 in DC migration [117]. ACKR4 is a scavenger receptor for CCL21 and CCL19, which in skin is expressed by keratinocytes and a subset of dermal LECs. Genetic deletion of Ackr4 led to an impaired migration of LCs and certain subsets of DCs to dLNs. Interestingly, this migration defect was rescued in animals double-deficient in both ackr4 and ccl19. Since DCs are known to start expressing CCL19 as they mature, the findings of this study suggested a role for keratinocyte-expressed ACKR4 in scavenging DC-produced CCL19, to allow the DC to better sense and migrate towards CCL21-expressing lymphatics [117].

#### 6.2.2. Other Chemotactic Cues

Although loss of CCR7 or blockade of CCL21 profoundly reduces DC migration to dLNs, also other chemotactic cues, have been described to contribute to this process [50,52] (Table 1). Under inflammatory conditions, LECs in afferent lymphatics upregulate various adhesion molecules and chemokines [52,74]. Two of them, namely CXCL12 and CX3CL1, which bind to DC-expressed CXCR4 and CX3CR1, were shown to regulate DC trafficking to dLNs in inflammation [118,119]. Intriguingly, we recently found the CXCL12 and CX3CL1 were preferentially upregulated in inflamed dermal collectors, rather than in capillaries [74]. Considering that CCL21 is expressed at lower levels in collectors as compared capillaries [59,74], this might indicate that CXCL12 and CXCL3 have a special role in supporting DC entry through collecting vessels in inflammation (Figure 3B, panel X).

Another chemoattractant that contributes to DC migration is the signaling phospholipid sphingosine-1-phosphate (S1P). S1P is present at high concentrations in blood and lymph but at low concentrations in tissues such as in LNs [57]. In peripheral tissues S1P is produced by LECs [120] and binds to the S1P receptors S1P1 and S1P3 expressed in maturing DCs [121,122]. Treatment with the S1P analog FTY720, a functional antagonist, or genetic deletion of S1p1 (but not of S1p3) resulted in impaired DC migration from the skin to dLNs. Interestingly, genetic deletion of S1p3 affected the migration of DCs from the intestine to dLN, revealing a fundamental role of S1P1 and a tissue-dependent contribution of S1P3 in DC migration [121,122].

#### 6.2.3. Adhesion Molecules

In contrast to leukocyte extravasation from blood vessels, which is a highly integrin-dependent process, DC migration into and through afferent lymphatics in steady-state conditions occurs independently of DC-expressed integrins [49]. This phenomenon is likely explained by the very low expression of integrin ligands ICAM-1 and VCAM-1 in LECs under steady-state conditions. Conversely, in inflammation, when LECs upregulate ICAM-1 and VCAM-1 [52,53,74], DC migration becomes integrin dependent. Blockade of LEC-expressed VCAM-1 and ICAM-1 or of DC-expressed LFA-1, or genetic deletion of the integrin subunit beta 1 (ITGB1—ligand of VCAM-1) was shown to diminish DC migration to dLNs in adoptive transfer and FITC painting experiments in mice [52,53,74,80,123,124]. Our group has recently reported that under inflammatory conditions, VCAM-1 is preferentially upregulated in LECs in collecting vessels as compared to LECs in capillaries [74]. In fact, our experiments identified VCAM-1 as a first molecule mediating entry of DCs directly into dermal collectors. By by-passing the slow active crawling step in lymphatic capillaries (Figure 3B, panel 2), this new entry route allowed for more rapid DC migration from peripheral tissues to dLNs (Figure 3B, panel X).

Besides the adhesion molecules ICAM-1 and VCAM-1, another well-described molecule that mediates DC trafficking to dLNs is the hyaluronan receptor LYVE-1, which is exclusively expressed in lymphatic capillaries [47]. A recent study showed that during inflammation the leukocyte receptor CD44 organizes the distribution of hyaluronan and regulates DC adhesion and transmigration through the lymphatic endothelium [125]. Further molecules reported to guide DC migration through afferent lymphatics are summarized in Table 1.

## 7. T Cell Migration through Afferent Lymphatics

### 7.1. T Cell Types and Functional Relevance of Migration

Since the 1960s, it has been well known that naïve lymphocytes are constantly recirculating between blood and SLOs [141,142,143]. Within LNs, naïve T cells sequentially interact with numerous DCs in search of antigen. If a T cell does not find its cognate antigen, it emigrates across cortical and medullary sinuses and exits the LN via the efferent LV. Conversely, upon antigen recognition, naïve T cells proliferate and, over the course of several days, differentiate into effector T cells (T_eff_) that egress from SLOs to access the inflamed peripheral tissue in order to fight the antigenic source (typically an infective agent). After elimination of the pathogen, most T_effs_ will die. However, some cells survive and develop into memory T cells, which will provide local and systemic defense in case of future pathogen exposures. [141,142].

Early studies identified two distinct memory T cell subsets, which can be distinguished based on their homing and effector properties [144]; namely, so-called central memory T cells (T_CM_) and effector memory T cells (T_EMs_). T_CM_ circulate exclusively between blood and SLOs and express adhesion molecules required for homing to LNs, such as CCR7 or L-selectin [141,142]. Conversely, T_EMs_ lack expression of LN homing molecules such as CCR7 and L-selectin but express adhesion molecules and chemokine receptors required for extravasation into peripheral tissues. More recently, further memory subsets, namely, tissue resident memory T cells (T_RM_) and recirculating memory T cells (T_RCM_), have been identified [41,145,146]. In contrast to T_EM_ and T_RM_, T_RCM_ express CCR7 and consequently may exit the tissue via migration into CCL21-expressing afferent lymphatics. From there, they migrate to dLNs and/or recirculate via blood to other peripheral sites. T_RCM_ have been identified both in mice [41] and in humans, where at least two subsets exist [147]. In contrast to T_RCM_, which provide extensive immune surveillance to distal tissues, the CCR7^-^ T_RM_ remain in the tissue and provide local and immediate protection upon reinfection with pathogens. It is likely that the different memory subsets work together, as there is some evidence suggesting that T_RM_ delay pathogen spread and will “sound the alarm” for the recruitment of circulating T_RCM_ [145]. However, the exact contribution of tissue-resident versus circulating T cells in pathogen clearance is still unclear, and in several experimental models of viral or bacterial infection, T_RM_ have even been shown to suffice for immune protection [146].

While T_RCM_ are thought to recirculate through tissues both in steady state and during inflammation, T_eff_ are mainly recruited to inflamed/infected tissues. T_eff_, therefore, expresses high levels of adhesion molecule and chemokine receptors required for homing to peripheral tissues, but are traditionally thought to be CCR7 negative, which would confine them to the first tissue they extravasate into. However, there are, meanwhile, several studies suggesting that T_eff_ in tissues retain low levels of CRR7 and that the ability to exit might even be beneficial for preventing overshooting inflammatory responses and for immune surveillance. For example, in a murine model of Morbus Crohn’, blocking CCR7 with antibodies or adoptive transfer of CCR7-deficient CD4^+^ T_eff_ exacerbated the disease symptoms [148]. Similar results were obtained in a delayed-type hypersensitivity (DTH) response model in the murine skin, where transgenic overexpression of CCR7 enhanced egress of antigen-specific CD4 T helper cells, thereby reducing tissue inflammation, whereas CCR7-deficiency diminished tissue egress and enhanced inflammation [149]. Similarly, in the context of viral infection in the lung, it was shown that lymphatic egress of T_eff_ depended on antigen recognition; in presence of cognate antigen, T_eff_ displayed reduced CCR7 responsiveness and consequently remained in the tissue, whereas non-specific T_eff_ retained CCR7 expression and egressed from the tissue [150,151]. It is likely that this type of “fine-tuning” of CCR7 levels helps to remove non-specific T cells form the inflammatory site, thereby preventing potential bystander T cell activation and immune pathology. Moreover, by egressing and returning to the blood, non-specific T_eff_ would get another chance to extravasate at another tissue site where their cognate antigen might be present.

Besides providing exit routes for non-specific bystander T_eff_ or for recirculating memory cells, T_reg_ also reportedly migrate via afferent LVs. In fact, several recent studies in mice found that T_reg_ represent the main T cell subset emigrating from sites of inflammation to dLNs [38,42,43]. T_reg_ that have migrated from the skin or from the colon via afferent LVs to dLNs display a more suppressive phenotype and function compared with T_reg_ that have entered LNs from the blood stream via high endothelial venules (HEVs) [38,42,43]. Similarly, studies in tissue allografts have also showed that T_reg_ migration to dLNs is required for efficient downregulation of the anti-allograft immune response [48,152,153]. Overall, these studies support the emerging concept that dampening overshooting immune responses and preventing autoimmunity is a key function of cellular traffic through afferent LVs.

However, with regard to T_reg_, it must be considered that, at present, there is only evidence from studies in mice regarding the striking frequency of T_reg_ amongst the cells migrating through afferent LVs [38]. The likely explanation for this is that at the time when most cannulation studies in humans/sheep were performed [14,16,17,36], the importance of T_reg_ and good markers for their identification were not yet known [154]. Thus, these studies still warrant repetition for a better understanding of the human relevance. Another important issue to consider when comparing findings made in mice and in humans is the fact that humans—who are exposed to many more microbes compared to mice housed under laboratory conditions—have by far greater memory T cell populations in blood and in peripheral tissues [155]. This adds an additional level of complexity when comparing memory T cell responses or the relative contribution of T_RM_ and T_RCM_ in the murine and the human systems.

### 7.2. Molecular Mechanism of T Cell Migration

#### 7.2.1. CCR7/CCL21-Axis

Similar to DCs, the CCR7-CCL21 axis represents the predominant pathway involved in T cell egress, including T_reg_ [156], from peripheral tissue via afferent LVs in steady state and in inflammation. Around 40–50% of all skin-associated CD4^+^ T cells in mice [41] and humans [147,157] express CCR7 and almost all migratory T cells are CCR7 positive [41,147]. Surprisingly, even though CCR7 is required for lymphatic egress during acute inflammation, recent studies indicated that there is no CCR7 dependency in chronic inflammation [158] and in tumor growth [159]. However, the latter finding contrasts to a previous report demonstrating a key role for migratory, CCR7-expressing migratory DCs in transporting tumor antigen and mediating the priming of CD8^+^ T cell responses in tumor-dLNs [160].

#### 7.2.2. Other Chemotactic Cues

As for DCs, S1P is important for T cell migration through afferent LVs to dLNs. Notably, S1P was originally identified to mediate egress of lymphocytes from LNs into efferent lymphatics [161]. In subsequent studies, S1P was found to also be important for T cell egress from nonlymphoid tissues [82,158,162]. Blocking of S1P signaling with FTY720 reduced T cell migration to LNs by causing accumulation of these cells near LVs [82,162]. In addition, CD4^+^ T_eff_ adoptively transferred into murine footpads displayed a defect in entry into afferent LVs when S1P1 and S1P4 were blocked. Interestingly, the same study also found that S1P signaling via LEC-expressed S1P2 impacted the junctional integrity of afferent LVs and also expression of the T cell trafficking molecule VCAM-1 [162]. Thus, S1P impacts trafficking by acting on both LECs and T cells.

Further evidence for the importance of S1P in T cell egress comes from the fact that T_RMs_ downregulate S1pr1 and S1pr5 that causes them to remain in the tissue [163,164]. Many T_RMs_, but not all, express CD69, which was shown to down-regulate S1P1. *Cd69^-/-^* T cells can enter the skin but do not form active T_RMs_ that are retained, whereas application of FTY720, which downmodulates S1P1 in T cells, rescues the effects of CD69-mediated retention in the skin [165,166].

#### 7.2.3. Adhesion Molecules

In addition to chemotactic signals, T cells also use various adhesion molecules for their migration through afferent lymphatics. LEC-expressed macrophage mannose receptor (MMR) interacts with the cell surface molecule CD44, which is expressed by most T cells found in afferent lymphatics [41,167,168]. The interaction of MMR with CD44 supports CD4^+^ and CD8^+^ T cell egress from skin, by increasing T cell adhesion to LVs [169,170]. Furthermore, the lymphatic endothelial and vascular receptor-1 (CLEVER-1), another protein expressed by both efferent and afferent LVs, was shown to be important for T cell entry into afferent LVs. Blockade of CLEVER-1 significantly decreased skin egress of CD4^+^ and CD8^+^ T cell to the dLN in vivo [171,172]. Interestingly, CLEVER-1 was recently also found to mediate DC migration to dLNs [137].

Similar to DCs, also lymphatic-expressed ICAM-1 and VCAM-1 are important for T cell migration to dLNs. A recent study from our lab using IVM showed that T cells crawling within lymphatic capillaries use LFA-1/ICAM-1 interactions to enhance their speed of migration under inflammatory conditions. Moreover, LFA-1/ICAM-1 interactions impacted T cell transmigration and the overall process of T cell migration from skin to dLNs [65]. In addition, VCAM-1 expressed by LECs was shown to mediate migration of T_reg_ but not of naïve CD4^+^ or CD8^+^ T cells from skin to dLNs [48]. Besides directly interacting with VCAM-1 during transmigration, T_reg_ were found to use lymphotoxin (LT) to trigger lymphotoxin-beta receptor (LTβR) signaling in LECs, thereby upregulating the expression of VCAM-1 and of other trafficking molecules such as CXCL12 and CCL21 in LECs [48,173]. By this means, T_reg_ were suggested to function as “gatekeepers” that control tissue exit of other leukocytes through afferent lymphatics, thereby supporting the resolution of inflammation [174].

With regard to the involvement of VCAM-1 in T_reg_ exit from inflamed tissues, it is interesting to note that VCAM-1 blockade also reduced the migration of adoptively transferred, in vitro generated CD4^+^ effector T cells from inflamed skin to dLNs [65]. Considering the recent findings from our group showing that VCAM-1 is preferentially upregulated in lymphatic collectors and has a role in DC migration through collecting vessels, these findings might suggest that T cells can also use this fast route to migrate to dLNs [74]. Table 2 provides a summary of molecules thus far shown to mediate T cell migration through afferent LVs.

## 8. Beyond Transport: Emerging Roles of Afferent Lymphatics in Immune-Modulation

Time-lapse imaging studies have revealed that DCs and T cells migrating to dLNs spend many hours crawling and arresting within lymphatic capillaries [21,59,62,65]. During this time, cells are in close contact with the endothelium, allowing for leukocytes and LECs to “communicate” and influence each other. As previously mentioned, the interaction of DCs with capillary LECs was shown to induce release of CCL21 [112], whereas T_reg_ interacting with LVs via LT/ LTβR axis where shown to induce upregulation of VCAM-1 and other trafficking molecules [48,173]. However, there is also evidence for opposite effects, in which molecules released or expressed by LECs in afferent LVs impact the immune-stimulatory phenotype of leukocytes. Over the last 10 years, several studies have shown that migratory DCs display a more tolerance-inducing phenotype and activity compared to LN-resident DCs [175,176]. So far, only few molecular interactions that mediate this effect have been identified: For example, the LEC-expressed adhesion molecule CLEVER-1 was not only shown to impact DC and T cell migration to dLNs, but also to suppress expression of costimulatory molecules in DCs. In absence of CLEVER-1, higher antigen-specific proliferative responses of T cells were observed in dLNs [137]. Similarly, another study identified LEC-derived prostaglandins as negative modulators of DC maturation [177]. Finally, a third study found that, in the absence of infection, interactions between LEC-expressed ICAM-1 and the DC-expressed integrin Mac-1 (CD11b/CD18) reduced expression levels of the costimulatory molecule CD86 in migrating DCs [178].

A similar picture of afferent lymphatics functioning as an immunosuppressive compartment for migrating leukocytes is also emerging for T cells. As already mentioned, several studies have reported that T_reg_ that migrate through afferent lymphatics are more suppressive then T_reg_ that have entered LNs trough HEVs [38,42,43]. However, no molecules involved in this process have been identified so far. By contrast, several studies have reported that LECs and other stromal cells in LNs express molecules that inhibit T cell activation [179,180]. Moreover, LN LECs also act as nonprofessional APCs, expressing MHC class I and II and PD-L1, but lacking expression of co-stimulatory molecules. In fact, LN LECs can present endogenous and exogenous antigens to T cells, thereby modulating their activation [181,182]. While most studies so far have identified immunosuppressive roles of antigen-presentation and T cell activation by LECs, it is worth mentioning that LN LECs were recently also found to efficiently prime effector-memory T cells [183]. Moreover, the capacity of LECs to take up and “archive” antigen for long time periods following viral challenge and vaccination was shown to positively influence protective immunity provided by T_CM_ [184]. Whether the same antigen-presenting functions and modes of immunosuppression or immune activation also occur in peripheral lymphatics is less well understood. However, several pieces of evidence in support of the latter have recently emerged, primarily in the context of tumor growth. Lane et al. recently described a role for interferon-gamma-induced PD-L-1 expression in peripheral lymphatics in reducing the activity of cytotoxic T cells [185]. The upregulation of PD-L1 by tumoral LECs was also reported by another study, which further showed that PD-L1 blockade resulted in increased T-cell activation by antigen-presenting LECs in vitro [186]. The immunosuppressive activity of tumoral lymphatics is further suggested by recent studies showing a striking correlation between lymphatic vessel density and the response to cancer immunotherapy both in murine models and in patients [187,188]. These studies further suggest that T cell functions in peripheral tissues are modulated by direct interactions with lymphatic vessels.

Besides serving as sites for immune-modulatory interactions with tissue-exiting and possibly also with tissue-resident T cells emerging evidence suggests that lymphatic capillaries also provide a compartment for DC-T cell interactions [21]. While activation of naïve T cells by DCs primarily occurs in SLOs, DCs are known to also interact with antigen-experienced effector/memory T cells in peripheral tissues such as the skin, but have thus far primarily been observed in the extra- or peri-vascular space [85,189]. Performing IVM during an ongoing DTH response in the murine ear skin, we observed that both cell types not only arrest on the endothelium or cluster within lymphatic capillaries, but also engaged into intralymphatic DC-T cell interactions. The length of these interactions was significantly increased in presence of cognate antigen recognition [21]. Intriguingly, only T cells but not DCs occasionally exited from the lumen back to the tissue. At present, we do not know whether such exit events were preceded by antigen recognition events, but it is intriguing to speculate that interactions within lymphatic capillaries might influence the T cell’s behavior to exit back into the tissue or to continue onwards to the dLN.

Considering that we presently know very little about the different cellular interactions occurring within or with afferent lymphatic capillaries, combined with the fact that the lymphatic network is extremely plastic and rapidly adapts during chronic inflammation, infection or tumor growth [2,3], it is likely that in the future many more immune-modulatory pathways and functions will be identified for this compartment.

## 9. Conclusions

Over the last decade, research on leukocyte trafficking through afferent lymphatics has made enormous progress thanks to the development of new techniques, such as the generation of numerous lymphatic-specific KO and reporter mice, time-lapse imaging or the use of photoconvertible mouse models that allow endogenous cell migration to be studied. While we nowadays have a good understanding of the general mechanisms of lymphatic trafficking, the migration specifics of different DC or T cell subsets through afferent lymphatics remain poorly understood. Similarly, less is known about the trafficking of other, particularly rarer, immune cell types such as gamma-delta T cells [190], NK cells [191] or innate lymphoid cells [192]. In the future, combinations of photoconversion studies with single-cell sequencing approaches are likely to provide new insights on the migration of such specific immune subsets and their implications in immunity.

Besides specific research, the field of lymphatic trafficking has also greatly profited from the increasing knowledge of the involvement of lymphatics in multiple body functions and diseases [2,3] and, most obviously, their contribution to immune regulation [181,182]. It is by now well known that the lymphatic vasculature presents different morphology and molecular signatures in different tissues [193]. Moreover, there is ample evidence showing that lymphatics respond and adapt in a stimulus and tissue-specific manner to inflammatory and immunologic challenges [52] and that LECs actively shape the immune response [179,181,182]. Therefore, more research will now be needed at the crossroads of lymphatic biology, immunology and leukocyte trafficking to better understand the pathogenesis of different diseases and develop novel therapeutic approaches that target the lymphatic vasculature. Enhancing or preventing tissue egress of a specific DC or T cells subset through afferent lymphatics could undoubtedly be of interest in the context of vaccination [194], transplant rejection [195] and also chronic inflammatory conditions. However, the fact that most trafficking molecules thus far identified in DC and T cell migration through lymphatics are, at the same time, well-known players in leukocyte extravasation from blood vessels, remains a key challenge at present. Therefore, for the development of new therapeutic approaches that exclusively target trafficking through afferent LVs, it will be crucial to identify molecules with specific expression on lymphatics but not in the blood vasculature.

## Figures and Tables

**Figure 1 cells-10-01269-f001:**
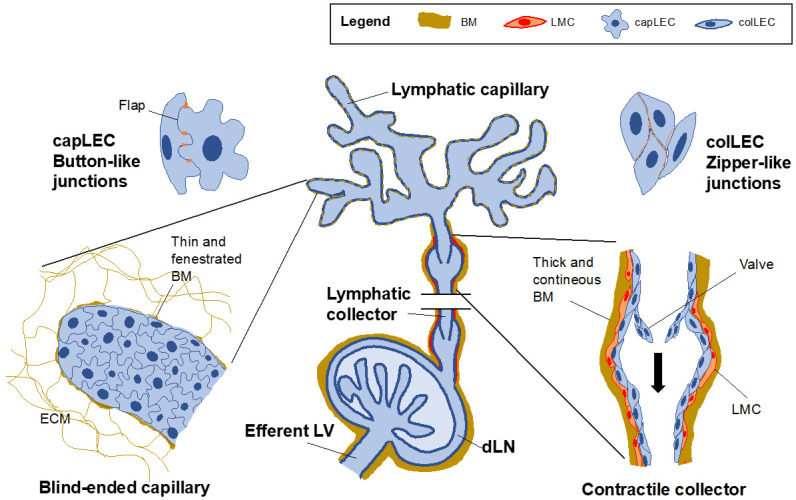
Morphological and anatomical characteristics of afferent lymphatics. Afferent LVs begin as blind-ended initial capillaries in peripheral tissues. Lymphatic capillaries have a wide lumen and are surrounded by a thin and fenestrated basement membrane (BM). LECs in lymphatic capillaries (capLECs) have a characteristic oak-leaf shape and are attached to each other by discontinuous button-like junctions, thereby generating open flaps (primary valves). capLECs are connected to the interstitial extracellular matrix (ECM) via anchoring filaments, allowing the flaps to open when interstitial fluid pressure increases. This characteristic setup, together with the discontinuous BM render initial capillaries specialized structures for fluid uptake and leukocyte intravasation. Lymphatic capillaries subsequently merge into lymphatic collectors, which eventually connect to dLN. Contrary to capillaries, LECs in collectors (colLECs) are elongated in the direction of flow and are attached to each other by continuous zipper-like junctions. Collecting vessels are also surrounded by a thick and continuous BM and covered by a layer of lymphatic muscle cells (LMCs). This setup, together with the presence of intralymphatic valves, allows lymphatic collectors to periodically contract and expand, thereby generating flow and propagating lymph in the direction of the dLN. Efferent LVs exit from LNs and connect with subsequent LNs to eventually merge into the thoracic and right lymphatic ducts.

**Figure 2 cells-10-01269-f002:**
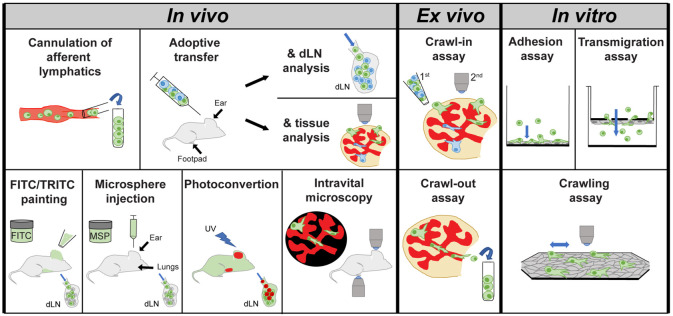
Schematic depiction of current in vivo and ex vivo or in vitro methods used to study leukocyte migration into and within afferent lymphatics. Detailed explanations and references to studies employing these methods are provided in the text of Section 4.

**Figure 3 cells-10-01269-f003:**
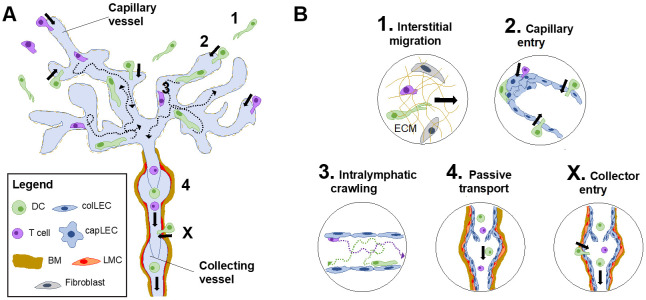
Leukocyte migration through afferent lymphatics occurs in a step-wise manner. (**A**) Summary of the key steps in DC and T cell migration from peripheral tissues through afferent LVs. (**B**) Description of the individual steps. *Step 1:* Interstitial migration: The interstitial space is comprised of fibroblasts and extracellular matrix (ECM) through which DC and T cells migrate in an ameboid fashion towards initial lymphatic capillaries. *Step 2:* Capillary entry: Attracted by the peri-lymphatic CCL21 chemokine gradient, DCs and T cells approach blind-ended capillaries and enter through specialized flaps formed by discontinuously joint capillary LECs (capLECs). *Step 3*: Intralymphatic crawling: DCs and T cells actively crawl and patrol within the capillary lumen, thereby interacting with the lymphatic endothelium. *Step 4:* Passive transport: migratory DCs and T cells eventually reach the downstream collecting vessels segments. Here, lymph flow increases due to vessel contractions mediated by the collector-surrounding lymphatic muscle cells (LMCs), leading to the detachment of leukocytes and their passive and rapid transport towards the dLN. *X*: Shortcut into afferent LVs: Under inflammatory conditions, DCs can additionally transmigrate and directly enter into lymphatic collectors [74].

**Table 1 cells-10-01269-t001:** Molecules involved in DC cell migration through afferent LVs.

Molecule	Comments	References
CCR7/CCL21/CCL19	*Ccr7* deletion and CCL21 blockade in mice severely compromises DC migration to dLNs CCL21 secreted by LECs in afferent capillaries and guides intralymphatic DCs in down-stream direction of the dLN *Ccl19*-deficient mice display no migration defect	[51,104] [59] [116]
ACKR4	Genetic deletion of *Ackr4* reduces DC migration to dLN. This defect is rescued in mice doubly deficient in *Ackr4* and *Ccl19*	[117]
CXCL12/CXCR4	CXCL12 and CXCR4 mediate cutaneous DC migration to dLNs	[119]
CX3CL1/CX3CL1R	CX3CL1 promotes DC migration from inflamed skin to dLNs.	[118]
S1P/S1P1/S1P3	Bone-marrow DCs migrate towards S1P. FTY720 treatment blocks DC migration from skin to dLNs. Endogenous DCs require S1P1 and S1P3 for migration from the intestine to dLNs, but only S1P1 for migration from skin to dLNs.	[121] [122]
Integrins (ICAM-1/VCAM-1)	DC migration to dLNs is integrin independent in the steady state but integrin dependent during episodes of inflammation. Loss of VCAM-1 in lymphatic collectors reduces rapid DC migration.	[49,53,123,124] [74]
Rho-associated protein kinase (ROCK)	Rock inhibition decreases intralymphatic crawling and overall DC migration to dLNs	[62]
L1CAM	Mice lacking L1CAM expression in endothelial cells display reduced DC migration to dLNs.	[126]
JAM-A/JAM-C	DC migration is increased in mice lacking JAM-A expression. Treatment of mice with a JAM-C blocking antibody enhances DCs migration and boosts immune responses.	[127] [128]
LYVE-1	LYVE-1 expressed in capillary LECs supports docking of DCs to LECs and migration to dLNs. CD44 controls the density of the hyaluronan glycocalyx, regulating the efficiency of DC trafficking to LNs.	[47] [125]
Podoplanin/CLEC-2	Reduced crawling on podoplanin positive vessels and reduced migration to dLNs in DCs lacking CLEC-2.	[129]
Semaphorin3a (Sema3a)	Sema3a promotes actomyosin contraction via its receptors Plexin-A1 and Neuropilin-1 (NRP1) and facilitates DC entry into afferent lymphatics and migration to dLNs.	[130]
Metalloproteases (MMP)	Blocking MMP-2 and MMP-9 reduces the migration of skin DCs to dLNs.	[131,132]
Prostaglandin-Receptors	DC migration to dLNs is increased after treatment with prostaglandin E2 by modulating CCR7 signaling and MMP-9 expression. PGE2 has a dose-dependent effect in regulating DC migration: High concentrations inhibited cell migration, whereas low concentrations exhibited the opposite effect	[132,133] [134]
CCR8/CCL1	Monocyte-derived DCs express CCR8 which regulates their migration to dLNs in inflammation.	[57,135]
Leukotriene B4 Leukotriene C4	Stimulation of LTB4 and LTC4 upregulates CCR7 and CCL19 in DCs and supports egress from skin to dLNs.	[136]
CLEVER-1	DC trafficking from the skin into the dLNs is compromised in the absence of CLEVER-1.	[137]
Migration inhibitory factor (MIF)	Autocrine and paracrine MIF activity acting via CD74 contributed to the recruitment of DCs to the dLNs.	[138]
ALCAM	DC migration to lung-dLNs is reduced in *Alcam-*deficient mice Blocking ALCAM leads to DC retention in corneal allografts, likely by prevent migration into lymphatics.	[56] [79]
Osteopontin (OPN)	LEC-expressed OPN supports DC migration to dLNs by interacting with CD44 and alpha v integrin	[139]
PD-L1	PD-L1 intracellular signaling controls DC migration from skin to dLNs by regulating CCR7-mediated chemotaxis.	[140]

**Table 2 cells-10-01269-t002:** Molecules involved in T cell migration through afferent LVs to dLNs.

Molecule	Comments	References
CCR7/CCL21	In mice, *Ccr7^−/−^* T cells display a profound reduction in migration from peripheral tissues to dLNs. In humans, all recirculating memory T cell subtypes are CCR7^+^	[41] [147]
S1P	Treatment with FTY720 reduced T cell migration to LNs Blocking of S1P1 and S1P4 reduce entry of T_eff_ CD4+ into afferent LVs S1P2 in LECs regulates T cell motility and transmigration	[82] [162] [162]
CD44/Macrophage mannose receptor (MMR)	Interaction of MMR in LECs with CD44 in T cells mediates CD4^+^ and CD8^+^ egress from skin	[169,170]
CLEVER-1	CLEVER-1 blockade reduces CD4^+^ and CD8^+^ T cell migration from the skin to the dLN	[171,172]
ICAM-1/VCAM-1	T cells require LFA-1/ICAM-1 interactions promoting T cell crawling and overall migration through afferent LVs T_reg_ migration to dLNs depends on VCAM-1 T_H1_ cell migration to dLNs depends on VCAM-1	[65] [65,152]
Lymphotoxin (LT)	Blockade of LTBR that binds to VCAM-1 reduced T_reg_ exit from the skin T_reg_ modulate LECs for transmigration of other cells, by stimulating LEC LTBR, to increase VCAM-1 and CCL21	[48]
CD69	CD69 downregulates S1P1, thereby inhibiting T cell egress from skin *Cd69^-/-^* T cells can enter the skin but do not form a T_RM_ population	[166] [165]
MECA-32 (PLVAP)	PLVAP expressed by LN LECs mediates lymphocyte entry across the subcapsular sinus into the LN parenchyma	[30]

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
