# Peer review of "Structure and Immune Function of Afferent Lymphatics and Their Mechanistic Contribution to Dendritic Cell and T Cell Trafficking"

_cells, 2021, doi:10.3390/cells10051269_

Round 1

Reviewer 1 Report

The review by Arasa, Collado-Diaz and Halin details the distinctive, specialized characteristics of afferent lymphatics and summarizes current knowledge of the cellular and molecular mechanisms by which DCs and T-cells migrate into and through these vessels. This manuscript also benefits from inclusion of a thorough and balanced evaluation of methods used to study lymphatic trafficking, and a discussion of the functional relevance of lymphatic migration of leukocyte subsets. On the whole, the review is well-written and provides an excellent overview for this area of research, citing a range of relevant and very recent papers in the literature. I have only a few minor comments:

  1. In Figure 2, the LEC monolayer should be shown plated on the undersurface of transwell inserts, as this is the approach used in the in vitro studies cited in references 51 and 80, to better reflect the basolateral-to-luminal transit of lymph migratory leukocytes
  2. Line 387. Authors should also cite Rigby et al 2015 (doi: 10.1189/jlb.1HI0415-149R) in addition to Hampton et al, to support the statement that neutrophils migrate more rapidly than DCs.
  3. A series of typos:

Typo line 346 “immunity”

348 “specialized” and “capturing”

353 “danger”

357 “molecules”

361 “from”

377 “immunostimulation”

395 “lymphatic”

411 “afferent”

420 “DCs”

421 “immunofluorescence microscopy” and “peri-“

424 “lymphatic”

426 “released”

428 “release”

442 “positively”

479 “dependent”

553, 555, 557 and 560 “non-specific”

563 References are in a different format that throughout the rest of the manuscript

580 This line should be revised as in its current form, it doesn’t make sense.

687 “through”

732 “endogenous”

733 “lymphatic”

755 “leukocyte”

757 “identify”

Author Response

The review by Arasa, Collado-Diaz and Halin details the distinctive, specialized characteristics of afferent lymphatics and summarizes current knowledge of the cellular and molecular mechanisms by which DCs and T-cells migrate into and through these vessels. This manuscript also benefits from inclusion of a thorough and balanced evaluation of methods used to study lymphatic trafficking, and a discussion of the functional relevance of lymphatic migration of leukocyte subsets. On the whole, the review is well-written and provides an excellent overview for this area of research, citing a range of relevant and very recent papers in the literature. I have only a few minor comments:

In Figure 2, the LEC monolayer should be shown plated on the undersurface of transwell inserts, as this is the approach used in the in vitro studies cited in references 51 and 80, to better reflect the basolateral-to-luminal transit of lymph migratory leukocytes

We thank the Reviewer the favorable assessment of our work and valuable comments. Following the Reviewer’s suggestion we have now changed the drawing in Figure 2 to showing the LEC monolayer on the lower side of the depicted transwell.

Line 387. Authors should also cite Rigby et al 2015 (doi: 10.1189/jlb.1HI0415-149R) in addition to Hampton et al, to support the statement that neutrophils migrate more rapidly than DCs.

Thank you for this comment and for reminding us about this well-known study, which had slipped our attention. It has now been added to the bibliography (cited in line 400).

A series of typos:

Typo line 346 “immunity”

348 “specialized” and “capturing”

353 “danger”

357 “molecules”

361 “from”

377 “immunostimulation”

395 “lymphatic”

411 “afferent”

420 “DCs”

421 “immunofluorescence microscopy” and “peri-“

424 “lymphatic”

426 “released”

428 “release”

442 “positively”

479 “dependent”

553, 555, 557 and 560 “non-specific”

563 References are in a different format that throughout the rest of the manuscript

580 This line should be revised as in its current form, it doesn’t make sense.

687 “through”

732 “endogenous”

733 “lymphatic”

755 “leukocyte”

757 “identify”

Thank you! – We have corrected these mistakes.

Reviewer 2 Report

Some suggestions to improve the review:

Lines 114-117:

In contrast to the old belief that large molecules are physically unable to enter the LNs, Kähäri et al. recently showed that large biomolecules can enter the LN parenchyma shortly after footpad injection (Kähäri et al., J Clin Invest 2019; Transcytosis route mediates rapid delivery of intact antibodies to draining lymph nodes.)

Lines 217-247:

The authors are listing different methods to study DC migration. Here it is also worth referencing the split ears assay developed by Lämmermann et al. (Lämmermann et al., Nature 2008; Rapid leukocyte migration by integrin-independent flowing and squeezing).

The authors also mention the in vitro transmigration assay but do not mention the similar mouse iv vivo experiment (Johnson et al., Nat Immunol 2017; Dendritic cells enter lymph vessels by hyaluronan- mediated docking to the endothelial receptor LYVE-1).

Lines 354-359 & 387-389:

In addition to the upregulation of co-stimulatory molecules, cytokine receptors and MHC expression, it would be helpful to briefly mention that DCs also reduce some of the adhesive molecules and functions to migrate into the LNs.

Lines 380-383:

The authors correctly point out the difference between resident and migratory DCs in T-cell priming. However, it would be worthful to briefly mention that these two subsets prime different T-cells. While migratory DCs mostly prime CD4+ T-cells in the draining LNs, resident DCs prime CD8+ T-cells in a delayed manner obtaining help from the previously activated CD4+ T-cells (Hor et al., Immunity 2015; Spatiotemporally Distinct Interactions with Dendritic Cell Subsets Facilitates CD4+ and CD8+ T Cell Activation to Localized Viral Infection).

Lines 684-692:

It is worth mentioning that LECs can pick up an antigen and archive it long after the initial infection (Tamburini et al., Nat Commun 2014; Antigen capture and archiving by lymphatic endothelial cells following vaccination or viral infection). Moreover, despite the lack of co-stimulatory molecules, it has been shown that LECs can efficiently prime memory effector T-cells. These LEC-primed memory T-cells can then efficiently become activated in response to the re-occurrence of the antigen (Vokali et al., Nat Commun 2020; Lymphatic endothelial cells prime naïve CD8+ T cells into memory cells under steady-state conditions).

Author Response

Some suggestions to improve the review:

Lines 114-117:

In contrast to the old belief that large molecules are physically unable to enter the LNs, Kähäri et al. recently showed that large biomolecules can enter the LN parenchyma shortly after footpad injection (Kähäri et al., J Clin Invest 2019; Transcytosis route mediates rapid delivery of intact antibodies to draining lymph nodes.)

We thank the Reviewer for making us aware of these important findings. We have now re-written the paragraph describing access of soluble molecules and cells into the LN parenchyma and also mention this study (p. 3, lines 116-130).

Lines 217-247:

The authors are listing different methods to study DC migration. Here it is also worth referencing the split ears assay developed by Lämmermann et al. (Lämmermann et al., Nature 2008; Rapid leukocyte migration by integrin-independent flowing and squeezing).

Please note the crawl-in experiments, developed by the group of Michael Sixt are mentioned in the text (p. 6, line 226) and shown in the top panel amongst the “ex vivo” experiments depicted in Figure 2.

The authors also mention the in vitro transmigration assay but do not mention the similar mouse in vivo experiment (Johnson et al., Nat Immunol 2017; Dendritic cells enter lymph vessels by hyaluronan- mediated docking to the endothelial receptor LYVE-1).

Sparked by the comment of the Reviewer we have modified the panel showing the adoptive transfer experimental setup to now integrate both the analysis of the dLN and of the injection site.  Moreover, we have inserted a sentence explaining this option in the corresponding text, now referencing two papers that performed this type of analysis (Johnson et al., Nat. Immunol. 2017 and Brinkmann et al., Nat. Comm. 20216) (p.5, lines 195-197). Since we needed more space for the adoptive transfer panel, we had to change the setup of Figure 2, now showing in the different type of experiments (in vivo, ex vivo or in vitro) in columns.

Lines 354-359 & 387-389:

In addition to the upregulation of co-stimulatory molecules, cytokine receptors and MHC expression, it would be helpful to briefly mention that DCs also reduce some of the adhesive molecules and functions to migrate into the LNs.

Unfortunately, it was not clear to us which adhesion molecules exactly the Reviewer is referring to. For example, LFA-1 and Mac1 appear to be equally expressed in immature and mature DCs, whereas ICAM-1 is an adhesion molecule that becomes upregulated (important for formation of the immunological synapse). Also other adhesion molecules like ALCAM and CD44 that are important for DC migration appear to remain expressed in DCs upon maturation. In the case of Langerhans cells (LCs), it is known that they downregulate E-cadherin upon leaving the epidermis, but surprisingly E-cadherin was shown to be dispensible for keeping LCs in the epidermis (Brand et al., JID 2020). Since we were not sure what we should add, we did not change anything on this point.

Lines 380-383:

The authors correctly point out the difference between resident and migratory DCs in T-cell priming. However, it would be worthful to briefly mention that these two subsets prime different T-cells. While migratory DCs mostly prime CD4+ T-cells in the draining LNs, resident DCs prime CD8+ T-cells in a delayed manner obtaining help from the previously activated CD4+ T-cells (Hor et al., Immunity 2015; Spatiotemporally Distinct Interactions with Dendritic Cell Subsets Facilitates CD4+ and CD8+ T Cell Activation to Localized Viral Infection).

We thank the Reviewer for this valuable suggestion and have now added two sentences mentioning these differences between migratory and resident DCs, including the suggested reference ( p. 10, lines 393-397).

Lines 684-692:

It is worth mentioning that LECs can pick up an antigen and archive it long after the initial infection (Tamburini et al., Nat Commun 2014; Antigen capture and archiving by lymphatic endothelial cells following vaccination or viral infection). Moreover, despite the lack of co-stimulatory molecules, it has been shown that LECs can efficiently prime memory effector T-cells. These LEC-primed memory T-cells can then efficiently become activated in response to the re-occurrence of the antigen (Vokali et al., Nat Commun 2020; Lymphatic endothelial cells prime naïve CD8+ T cells into memory cells under steady-state conditions).

Following Reviewer’s comment we are now also mentioning how antigen presentation or antigen archiving by LECs supports induction of protective immunity (p. 18, lines 711-717).
